# Streaming to Connect: Exploring How Social Connectedness Relates to Empathy Types and Physiological States in Remote Virtual Audiences

**DOI:** 10.3390/s25030872

**Published:** 2025-01-31

**Authors:** Katherine Wang, Jitesh Joshi, Youngjun Cho

**Affiliations:** 1Division of Psychology and Language Sciences, University College London, London WC1E 6BT, UK; katherine.wang.19@ucl.ac.uk; 2Department of Computer Science, University College London, London WC1E 6BT, UK; jitesh.joshi.20@ucl.ac.uk

**Keywords:** physiological sensing, HRV, empathy, vmHRV, virtual interaction, remote interaction, social connection

## Abstract

Examining remote virtual audiences is a vital part of understanding social experiences in modern human–computer interaction contexts. Doing so raises intriguing questions about how these mediated connections relate to emotional and physiological states. The vagus nerve is central to socioemotional processing and physiological well-being, with its activation represented as vagally mediated heart rate variability (vmHRV). We examine how participants’ social connectedness to virtual partners relates to their experience of socioemotional competences and psychophysiological states while observing streamed gameplay. In this experimental study with 48 participants, we compared self-reported empathy, empathic concern, and continuously measured vmHRV (from a PPG sensor) during different types of gameplay. The results revealed that viewers who felt greater social connectedness to remote partners also felt more empathic concern (quantitative detail) and had significantly heightened vmHRV (quant detail) across all conditions compared to those who felt lower connectedness. These findings reveal that stronger feelings of connectedness to remote partners are associated with enhanced socioemotional competences and physiological well-being. This research highlights the intertwined nature of social connectedness, empathy, and physiological health, providing valuable insights for designing virtual platforms that foster deeper interpersonal connections and promote well-being.

## 1. Introduction

The emergence of virtual platforms has transformed how people engage in remote, shared experiences, fostering novel opportunities for social connections. From live performances to online gaming and streamed events, virtual audiences watch real-time activities despite being physically distant. While the field of human–computer interaction has investigated the technologies that have reshaped traditional social interactions and revealed new ways of engaging remote audiences, concerns remain about their implications on socioemotional competence and support for social connectedness, particularly as remote audiences often lack the physical presence and nonverbal cues essential for fostering deep connections. An alarming trend that has been observed over the past decades has been a decrease in empathy, particularly in empathic concern (the concern for another’s well-being) [1]. As empathy is highly related to prosocial outcomes, its decline has unsurprisingly coincided with a rise in antisocial traits like narcissism and self-centeredness [2].

Socioemotional challenges brought about by the circumstances of the COVID-19 pandemic have had repercussions on declining empathy [3,4]. Empathic concern is central to meaningful interpersonal relationships [5]. Individuals with higher empathic concern are more likely to exhibit greater emotional sensitivity toward others and more likely to engage in acts of compassion [6,7]. These competences are closely tied to physiological processes [8], particularly the vagus nerve, which regulates emotional responses and social engagement, and are explained in the following section. As such, understanding how empathy relates to other prosocial competences or physiological states becomes paramount and can be better utilized to nurture empathic interactions.

Although prior research has explored the role of vagal tone in empathy and emotional regulation [9], its application in remote, virtual audience settings remains underexplored. The unique dynamics of remote audiences pose important questions about how social connectedness, empathy, and physiological states manifest in these contexts. While remote virtual interactions facilitate social connectedness, the quality of these interactions varies widely, often influenced by the perceived closeness or social connectedness between individuals. We focus on exploring the context of watching a remotely streamed game, as live stream gaming has become a prominent medium for fostering and supporting social connections and community-driven behaviors between gamers and audience members. We examine how social connectedness relates to different aspects of empathy and psychophysiological states in audiences of remotely streamed gameplay. Understanding these relationships is essential for designing virtual platforms and interactions that foster genuine connections and improve well-being. Moreover, little is known about how watching different scenarios, such as goal-oriented versus passive activities, may shape the relationship between social connectedness and emotional or physical outcomes.

This study investigates how social connectedness with a virtual partner influences viewers’ socioemotional competences and psychophysiological states during streamed gameplay. By integrating continuous physiological measurements with self-reported metrics, we aim to explore the connection between empathic concern, physiological synchrony, and the type of content being streamed. Our findings contribute to the growing body of research on how socioemotional states relate to physiological states and highlight the potential for using physiological sensing to better understand social experiences in virtual scenarios. More specifically, this study examines the following:The relationship between individuals’ perceived social connectedness, socioemotional competences (e.g., empathy and concern), and physiological states.How does the context of an interaction influence this relationship, particularly the effects of watching goal-oriented versus passive gameplay.

## 2. Related Work

### 2.1. Socioemotional Competences and Physiological States

Empathy, empathic concern, and compassion are among the distinct socioemotional competences presented in the domains and manifestations of socioemotional competences (DOMASEC) model [10]. Amidst the nuanced interpretations surrounding these terms, we distinguish the perspective-taking form of empathy (referred to in this paper as *affective empathy*, or simply *empathy*) from the sympathizing form (referred to in this paper as *empathic concern*, or simply *concern*) [7]. Related research also intertwines the concept of compassion, which encompasses the desire to help others and the behavior of doing so [11]. This section further explains these distinctions and introduces how they relate to physiological states.

Affective empathy entails a top-down process whereby perceiving another’s emotional state generates an affective state more relevant to the other’s situation than one’s own position [12,13,14]. As such, perspective-taking and personal distress are commonly measured as facets of affective empathy. Perspective-taking involves the cognitive effort to experience a situation from another’s point of view. Personal distress refers to the negative affective response experienced in reaction to the vicarious experience of another person’s distressed emotional or physical state and occurs when an individual merges the other’s plight with their own [15,16]. Indeed, studies have shown that the ability to recognize emotional expressions is closely linked to empathy levels [17].

Empathic concern emphasizes reacting and anticipating another’s emotions or contextual situation with an affective state coupled with regard for the other person’s well-being [18]. Here, the affective state occurs independently, unlike the vicarious feelings associated with affective empathy. Neuroscientific research revealed that empathic concern relates to the activation of socioemotional recognition and social motivation regions (i.e., the anterior insula and anterior cingulate cortex) rather than the regions related to physical pain when witnessing another’s physical distress and occurs in both intimate partners [19] and strangers [20]. Other research has proposed that the degree of social connectedness with another person closely relates to levels of empathic concern felt for them [21,22].

Physiological activity can offer valuable insight into a person’s experiences of socioemotional competences [8]. Central to socioemotional engagement is the vagus nerve, which activates when positive emotions are felt during social interactions [23]. Vagally-mediated heart rate variability (vmHRV) can possibly reflect this activation and indicate arousal of the parasympathetic nervous system [24,25]. There is evidence that increases in vmHRV are strongly related to socioemotional competences [25]. Likewise, vmHRV can be used as evidence of socioemotional competences during social interactions [9,26]. Prosocial emotions, such as those affiliated with feeling concerned for another, can lead to encouraging vagus nerve activation, depicted as increased vmHRV [27]. Heightened vmHRV has also been found in individuals with higher self-reported empathy [24]. Increases in vmHRV are more closely related to emotions derived from cognitive empathy than emotions not from appraising others’ distress [25]. Likewise, decreases in vmHRV have been found in social interactions where self-interest may take precedence over the well-being of others [26].

Among the socioemotional competences, compassion is the most commonly studied concerning vmHRV. Compassionate individuals consistently exhibited increased vmHRV [28,29], and research indicates that engaging in compassionate behavior can also lead to a persistent increase in vmHRV [25,30,31]. These findings highlight the crucial role socioemotional competences play in regulating vmHRV and how they can benefit overall physical well-being.

### 2.2. Social Connectedness and Shared Physiological States

One notable concept in the research of social connectedness is physiological synchrony, which refers to the dynamic and temporal co-ordination of physiological processes between people engaged in an interaction or shared experience [8,32]. Physiological synchrony has been documented among individuals in various autonomic responses, including heart rate, skin conductance, respiration, and temperature, across various relationships and social situations. For instance, romantic partners have shown similarities in their HRV patterns during positive romantic interactions, as well as during instances of relationship conflicts [33]. As vmHRV and other physiological indices are strongly linked to social affect, the alignment of these indices among individuals may suggest emotional convergence during a shared emotional experience of a specific variable or task [34]. Occurrences of physiological synchrony during social interactions have also been observed between co-workers [35], psychotherapists and clients [36], and mothers and infants [32].

Some research emphasizes that sharing physiological states is more closely tied to feelings of social connectedness [37], which is cultivated through shared experiences related to a particular event—such as a meeting—or an activity, such as a ritual. Still, other research proposes that physiological synchrony can occur among individuals regardless of the level of interpersonal familiarity or degree of social interaction [38,39,40]. For instance, merely witnessing another person experience physical pain triggers similar changes in neural activity and skin conductance in people, as if they were experiencing the same physical distress, irrespective of whether the other person is a romantic partner or stranger [19,41]. These changes also occur in response to observing another’s emotional pain. Cortisol responses to perceiving another’s psychological distress have been shown to occur in both romantic partners and strangers, regardless of gender or physical proximity [30] [42]. Evidence from the same study also found a positive correlation between these physiological changes and empathic concern, suggesting individuals who experience feeling more concerned for another may exhibit a more pronounced physiological response to perceiving others in emotional distress. For instance, observing their own child in a state of mental distress elicited more robust changes in the facial thermal patterns of mothers, which were also found to be more closely synchronized with their children compared to other children [43]. Similarly, the ability to accurately infer another person’s emotional state is associated with a higher frequency of participants’ HR measures correlating with those of the other person [44].

This highlights the significance of understanding the relationship between social connectedness, socioemotional competences, and physiological responses that can bypass the nature of existing relationships, social dynamics, and physical proximity. Although the observations of shared socioemotional and physiological states from this section have primarily been derived from collocated pairs of participants [9], it is worth considering how these findings translate to remote pairs in virtual scenarios that may be limited in the same social cues as those in face-to-face interactions. Therefore, in addition to examining the experiences of socioemotional competences and physiological states of viewers according to their perceived social connectedness to virtual partners, we also explore the degree to which their physiological activity synchronizes with these partners during remote gameplay sessions.

By building upon these studies, our research extends the application of vmHRV measurement to remote virtual audience settings. This adaptation of established methodologies allows us to explore how social connectedness and empathy manifest in digital interactions, providing valuable insights for designing virtual platforms that promote socioemotional well-being.

The study presented in this paper combines subjective measures of empathy with objective physiological data (vmHRV), offering a comprehensive analysis of how social connectedness relates to both emotional and physical well-being by answering the research question, “How does feeling social connectedness to a virtual partner relate to empathic and psychophysiological states of viewers?” By using continuous physiological measurements and self-reported metrics, our study aims to offer insights into how social connectedness to virtual partners is inextricably linked to socioemotional competences and physiological well-being in audiences. Additionally, the study explores the occurrence of shared physiological states between viewers and their virtual partner with viewers’ feelings of closeness towards the partner during virtual gameplay and its implications on social connectedness [45].

## 3. Methodology

Our study design leverages the established relationship between vmHRV and socioemotional competences to explore these dynamics in a novel context. By building on evidence from prior research (e.g., [24,46]), we employed continuous measurement of vmHRV using a PPG sensor to objectively assess participants’ physiological states alongside measuring empathy during remote gaming sessions. We adapted our environmental setup and approach from a previous study on physiological measurements and empathy evaluations during video viewing [47]. The following subsections detail our methodology.

### 3.1. Participants

The participants were enlisted from a research recruitment platform and local university research communities. In an effort to recruit pairs with varying degrees of social connectedness, two advertisements separately targeted pairs of strangers or romantic couples and close friends. This distinction was crucial to ensure that one group had a higher level of closeness from intimate relationships. A total of 48 individuals (median age ($M_{age}$): 28.0 years, minimum age ($min_{age}$): 18, and maximum age ($max_{age}$): 59) participated as 11 pairs of strangers and 13 familiar pairs (32 females and 16 males; viewers: 18 females and 6 males). Before beginning the study, all participants received an information sheet and provided informed consent. All participants indicated they felt mentally and physically healthy during participation and were compensated with a £15 Amazon voucher for 60 min at the end of the study.

### 3.2. Materials and Setup

The participants were placed in two separate rooms to simulate a remote gameplay scenario. In the viewer room, a first-person view of their partner’s perspective was broadcast onto a 67″ × 37″ 2D screen (bottom left image in Figure 1). Speech and audio communication between the viewer and their partner were enabled using the microphone and speakers on the HMD and PC, respectively, as suggested in a pilot of this study and as reported in previous study setups [48].

The setup for the virtual partner, referred to as the gameplayer from this point on, took place in a separate room to simulate a remote interaction. A VR wireloop game (top left image in Figure 1) purchased from the Steam storefront was run on a Meta Quest2 using AirLink. By using a controller, gameplayers dragged a white ring over differently shaped wires. The game was chosen due to its similarities to the Mirror-Tracing Persistence Task (MTPT) [5], a visual motor task that has been shown to induce psychological stress and elevate heart rate [49]. Compared to other commonly used laboratory stressor tasks (e.g., nonverbal math, Stroop Color-Word interference [50]), the MTPT engages goal-oriented performance [51] and takes advantage of VR spatial affordances, making it a more engaging gameplay task. Furthermore, VR elicits stronger emotional responses without being more difficult to play than a desktop video game [52].

#### 3.2.1. Physiological Sensing Setup and Data Collection

The primary physiological phenomenon we focus on in this study is heart rate variability (HRV). For this, we mainly used ear photoplethysmography (PPG) for HRV extraction (which is also referred to as pulse rate variability). The ear PPG sensor was specifically chosen due to its minimal interference with participant movements in the VR environment, avoiding common issues such as motion artifacts that can arise from hand- or wrist-worn devices. Other physiological measures, such as skin conductance, respiration rate, or face temperature, were excluded as they typically require sensor placements that could interfere with the gameplayer’s natural use of VR controllers and often cause inaccurate measurements [53]. We found this setup to be the least intrusive way to collect physiological data while preserving the ecological validity of the VR experience.

As this study required simultaneously collecting physiological data from two remote participants, we used PhysioKit, a physiological computing toolkit that distinctly supports synchronous raw data acquisition from multiple participants [53]. We used PPG sensors on both viewers and gameplayers to acquire blood volume pulse signals (BVP) and compute HRV metrics (Although we use the general term heart rate variability (HRV), we mainly refer to PPG-derived HRV, which is also called pulse rate variability (PRV) [54]). To minimize data loss from motion artifacts, one sensor was fitted on a finger of each participant’s nondominant hand, and a second one was clipped to their earlobe. Figure 2 depicts the setup used for gathering physiological data from the viewer. The sampling rate was set to 250 samples per second.

#### 3.2.2. Subjective Questionnaire

The participants filled out a short questionnaire immediately following each condition to measure the experiences of perceived social connectedness and state empathy during each experimental condition. The first question asked viewers how socially close they felt to the gameplayers (“*How close did you feel to the other person?*”). The subsequent questions were adapted from the Interpersonal Reactivity Index (IRI [55]), which measures dispositional empathy by using subscales. The empathic concern subscale from the IRI measures warmth, compassion, and concern for others and features items such as “*When I see someone being taken advantage of, I feel kind of protective towards them*”. Meanwhile, the perspective-taking subscale gauges the inclination to adopt others’ perspectives in daily life: “*I sometimes try to understand my friends better by imagining how things look from their perspective*”. Additionally, the personal distress subscale evaluates one’s own unease and discomfort in response to others’ emotions, with items like “*In emergency situations, I feel apprehensive and ill-at-ease*”. By drawing from these IRI subscales, the questions assessed empathic concern by asking, “*How concerned were you for your partner’s feelings?*”. As affective empathy is characterized by an individual’s distress in conjunction with their perception of another’s distress, we asked questions based on the perspective-taking and personal distress subscales, such as “*How stressed do you think your partner felt?”* and “*How stressed did you feel?*”. The participants rated their responses using a Visual Analogue Scale (VAS scale; [56]) ranging from 0 (“Not at all”) to 100 (“Very much”), which has been widely used as a valid and reliable assessment of acute pain intensity [56]. The adapted measures focus on assessing state empathy, capturing the viewers’ feelings at a specific moment during the experimental conditions.

#### 3.2.3. Open-Ended Questions

The participants were invited to answer three written free-form questions at the end of the study to contextualize the effects of the experimental conditions (which will be explained in a later section). The first two questions asked participants to reflect on what aspects contributed to their sense of closeness with the gameplayer (*“What aspects helped you feel connected to the gameplayer during this session?”*) and what caused them the most stress (*“What made you feel the most stressed while watching this gameplay session?”*). The third question invited them to suggest features they thought might encourage social connectedness (“What do you think could help you feel more connected to the gameplayer?”). We derived themes and sub-themes by using inductive thematic analysis [57] with NVivo 12 software [58]. Initially, codes were established by highlighting similar wording or semantic patterns and iteratively processed until themes emerged.

### 3.3. Procedure

The participants were randomly assigned the role of gameplayer or viewer and guided into separate rooms where they provided informed consent. Next, they completed a pre-test subjective questionnaire, which included questions on demographics and the subjective ratings mentioned above. Then, the participants received verbal instructions on the procedure and were fitted with physiological sensors. After measuring their resting baseline HRV for 5 min [22], the gameplayer was provided with instructions on how to play the game. Once they were fitted with the VR headset, they were given a chance to familiarize themselves with the game. To keep the viewer engaged during the session, we suggested they count the gameplayer’s in-game errors [59]. Both participants engaged in three counterbalanced conditions of gameplay, each lasting 5 min. We referred to these conditions as *ActiveEasy*, *ActiveHard*, and *Passive*, and we elaborate on the development of these tasks in the following subsection. The wireloop game was played during both *ActiveEasy* and *ActiveHard* conditions. The sound of a ticking clock and audio notifications counting down time were introduced during *ActiveHard* to induce stress through time pressure [60,61]. In asymmetric VR gameplay, the sound of ticking has also been shown to increase stress in both players and viewers, even if the sound’s visual source is obscured [62]. During the *Passive* condition, gameplayers were instructed to do as they pleased for 5 min as long as they remained on the game’s homescreen, as shown in the bottom right image of Figure 1. The participants were asked to fill out the same subjective questionnaire after each condition. The experimental protocol for this study is illustrated in Figure 3 and was approved by the University College London Interaction Centre ethics committee (ID Number: UCLIC/1920/006/Staff/Cho).

#### Task Design

The conditions in this study were designed to facilitate the investigation of contrasting gameplay types. The two *Active* conditions in this study (*ActiveEasy* and *ActiveHard*) exemplify simplified games of emergence because they present an open gameplay structure, potentially resulting in emergent novelty in gameplay, interactions, and sense-making [63,64]. The *ActiveHard* condition was deliberately designed to induce stress, making it the more challenging gameplay. The elements that contribute to the stressful nature of the task, such as the sound of a ticking clock counting down the time, the complexity of the wire shape, and the goal of completing the wire as many times as possible, give this condition a greater difficulty level that intends to evoke stress in players and viewers. On the other hand, while the gameplayers still encountered a ring and a wire in the *ActiveEasy* condition, the gameplay was more open in the sense that there was no defined goal in this condition. Furthermore, the wire was comparatively less complex than the one in the *ActiveHard* condition, making it easier to complete and allowing players to complete it at a leisurely pace, repeat it as desired, and explore different techniques. This openness consequently affects the viewer’s sense-making process, who might rely on different techniques to interpret the player’s behaviors and emotions. On the other hand, the *Passive* condition was more of an open-ended scenario because there are various possibilities for socioemotional interactions, but they are influenced by mechanics independent of the gameplay [65]. While the *Passive* condition also invited novel interactions, it was distinct from the gameplay in the *Active* conditions since the openness in this condition relies solely on social interactions. These contrasting conditions highlight how viewers’ social experiences and physiological responses vary with their sense of connectedness to players and the type of gameplay.

### 3.4. Analysis

#### 3.4.1. Physiological Synchrony

The preprocessing of BVP signals involved the application of a bandpass filter set within the frequency range of 0.7 Hz to 4.0 Hz. Subsequent to this, an evaluation of signal quality was conducted with the intent to eliminate segments compromised by motion artifacts. Given the highly periodic nature of BVP signals in healthy individuals, the measurement of auto-correlation can be a pivotal tool in identifying motion artifacts, which are characterized by substantial reductions in auto-correlation values [66]. Consequently, signal quality can be assessed by determining the maximum auto-correlation amplitude present within a normalized correlogram [66]. For this purpose, normalized correlograms were computed over intervals of 2.5 s, with a step interval of 0.5 s, thereby yielding a temporal resolution of 0.5 s to assess signal quality.

In order to estimate physiological synchrony, we computed the vagally mediated heart rate variability (vmHRV), which is extensively acknowledged as a dependable indicator of parasympathetic activity [25] over short-term periods ranging from seconds to minutes [67]. Among the various vmHRV metrics, the pNN50 was employed, calculated as the proportion of successive heartbeat intervals that differ by more than 50 ms.

To reliably compute pNN50, we used a 120-s window length with a 5-s step interval. The signal quality scores were averaged for each 120-s window, and segments with a score of less than 0.7 were discarded [68]. For each 120-s window discarded for one participant, the corresponding window of the respective partner was also discarded to ensure a balanced analysis between HRV metrics from time-matched windows. The pNN50 metrics were computed only for the good-quality time-matched segments. For the instances where the number of good-quality time-matched 120-s segments was less than three, the entire session of an experimental condition for a pair was dropped from further analysis.

For each experimental condition, we employed time-matched pNN50 vectors, which were derived from cleaned 120-s overlapping segments of paired streams and viewers, to estimate physiological synchrony. Specifically, we computed the Pearson correlation coefficient (*r*) [69] between paired pNN50 vectors. Figure 4 illustrates instances of both synchronous and nonsynchronous vmHRV signals observed between a gameplayer and a viewer across two distinct pairs.

#### 3.4.2. Subjective Ratings

Viewers were categorized into two groups based on their subjective ratings of social closeness compared to the median closeness rating for each condition. We refer to these groups as *HIGH* and *LOW*, depending on whether ratings fell above or below the median. We performed statistical analyses to examine and compare these two groups across experimental conditions using SPSS software (version 29.0.0.0 [70]). Given the ordinal nature of VAS data, we used a Spearman rank correlation (rs) to examine relationships between Closeness and other EmpathicConcern measures. As previously explained, AffectiveEmpathy was assessed by correlating PerspectiveTaking and PersonalDistress. Due to the lack of normality in the dataset, Mann-Whitney U tests were used as a nonparametric alternative to the independent samples *t*-test to examine differences in EmpathicConcern and vmHRV between groups across three experimental conditions. For similar reasons of non-normally distributed data, Wilcoxon signed-rank tests were conducted to assess variances in dependent variables within each Closeness group. Afterwards, Bonferroni post-hoc corrections were applied. Additionally, effect sizes (*r*) were calculated using $r=\frac{Z}{\sqrt{N}}$ [71]. These were reported based on Cohen’s criteria [72], with 0.10 indicating a small effect, 0.30 indicating a medium effect, and 0.50 indicating a large effect. The results of these analyses are presented in the next section.

## 4. Results

### 4.1. Closeness

As previously mentioned, the pairs were split evenly into two groups based on whether they fell above (*HIGH* group) or below (*LOW* group) the viewer’s median closeness rating (VAS) for each condition. Here, we elaborate on how the *HIGH* and *LOW* groups were defined for each condition. With the exception of one or two pairs, the viewers from familiar pairs had generally higher perceived closeness ratings than unfamiliar pairs. The average closeness ratings for viewers from familiar pairs (*N* = 13) were higher for the *ActiveHard* condition (*Mean* = 75.77; *SD* = 14.95) compared to the *ActiveEasy* condition (*Mean* = 63.7; *SD* = 19.96). On the other hand, viewers from unfamiliar pairs (*N* = 11) had higher average closeness ratings during the *ActiveEasy* condition (*Mean* = 33.36; *SD* = 29.33) compared to the *ActiveHard* condition (*Mean* = 28.45; *SD* = 7.55). We found this approach of grouping pairs based on median closeness ratings rather than familiarity was the best way to ensure that the results were more reflective of the emergent patterns of social connectedness rather than solely on pre-existing relationships. The formation of these groups for each condition can be found in Table 1. By applying this approach to each condition, we were able to better understand the influence of gameplay type on the emergence of social closeness.

### 4.2. Stress

Next, we wanted to establish the validity of our task design, which invoked different types of emotions experienced while watching gameplay, namely psychological stress. In order to confirm that the *ActiveHard* condition provided the most psychologically stressful gameplay for gameplayers of the three experimental conditions, we first conducted a Wilcoxon signed-rank test with Bonferroni correction (significance at 0.017), comparing it with the other conditions. As illustrated in Figure 5, gameplayers did, indeed, find the *ActiveHard* condition (*HIGH*: *Mdn* = 65.50; *LOW*: *Mdn* = 54.50) significantly more stressful than the *Passive* condition (*HIGH*: *Mdn* = 6.00; *LOW*: *Mdn* = 4.50). (*HIGH*: *Z* = −2.982, *p* = 0.003, −0.862; *LOW*: *Z* = −2.511, *p* = 0.012, *r* = −0.726). As intended, gameplayers also reported feeling more stressed during the *ActiveHard* condition than the *ActiveEasy* condition (*HIGH*: *Mdn* = 27.00; *LOW*: *Mdn* = 36.00). This was particularly significant for the *HIGH* group (*Z* = −2.981, *p* = 0.003), which had a large effect size (*r* = −0.862).

Viewers also found the *ActiveHard* gameplay significantly more stressful to watch than the *Passive* condition (*HIGH*: *Z* = −3.061, *p* = 0.002, −0.885; *LOW*: *Z* = −2.746, *p* = 0.006, *r* = −0.794). They also tended to perceive that it was more stressful for players than the *ActiveEasy* condition, but this was only significant for the *HIGH* group (*Z* = −2.590, *p* = 0.010), for which there was a large effect size (*r* = −0.749).

### 4.3. Empathic Concern

To assess how feeling socially connected to gameplayers (Closeness) related to how concerned viewers felt for them (EmpathicConcern), we performed a Mann-Whitney U test with a Bonferroni correction (*p* < 0.017). As illustrated in Figure 6, closeness elicited a significant difference in viewers’ EmpathicConcern ratings for all conditions with a large effect size: *ActiveEasy*, *U* = 26.50, *p* = 0.009 *r* = −0.536; *ActiveHard*, *U* = 7.50, *p* ≤ 0.001, *r* = −0.761; *Passive*, *U* = 20.00, *p* = 0.003, *r* = −0.614. While viewers from both the *HIGH* and *LOW* groups had higher EmpathicConcern ratings during Active conditions, these ratings did not differ significantly between the different conditions. While viewers from both the *HIGH* and *LOW* groups had higher EmpathicConcern ratings during the Active conditions, these ratings did not differ significantly between the different types of gameplay.

### 4.4. Affective Empathy

As previously mentioned, affective empathy can be described as the vicarious experience of another’s emotions. As such, viewers’ affective empathy was analyzed as a positive Pearson correlation coefficient (*r*), representing the relationship between viewers’ perspective-taking abilities in understanding how stressed players felt (PerceivedStress) and how much personal distress they experienced themselves (OwnStress), as shown in Figure 7. Affective empathy was strong and statistically significant during Active conditions for both the HIGH (*ActiveHard*, *r* = 0.696, *p* = 0.012; *ActiveEasy*, *r* = 0.865, *p* ≤ 0.001) and *LOW* viewers (*ActiveHard*, *r* = 0.621, *p* = 0.031; *ActiveEasy*, *r* = 0.806, *p* = 0.002). Table 2 depicts all correlation values from this analysis.

### 4.5. Psychophysiological States

#### 4.5.1. Between-Group Analysis

A Mann-Whitney U test with a Bonferroni correction (*p* < 0.017) showed that closeness elicited a significant difference in viewers’ HRV ratings for all conditions as illustrated in Figure 8: *ActiveEasy*: *U* = 67,968.00, *Z* = −7.963, <0.001, *r* = −0.267; *ActiveHard*: *U* = 46,742.00, *Z* = −12.000, <0.001, *r* = −0.415; Passive: *U* = 67,206.00, *Z* = −6.371, <0.001, *r* = −0.219.

#### 4.5.2. Within-Group Analysis

Post-hoc analyses using Wilcoxon signed-rank tests were conducted with a Bonferroni correction (*p* < 0.008) to determine the differences in HRV between the experimental conditions for each group. There was a statistically significant difference in viewers’ vmHRV depending on the difficulty of the conditions (Table 3). Viewers from the *HIGH* group had significantly higher parasympathetic activity during Active conditions compared to the *Passive* condition. On the other hand, viewers from the *LOW* group had significantly lower HRV during Active conditions compared to the *Passive* condition.

More specifically, when comparing their vmHRV during the *Passive* condition (*Mdn* = 30.59), the HIGH viewers had a higher vmHRV during the *ActiveEasy* (*Mdn* = 44.44, *Z* = −4.075, <0.001, *r* = −0.194) and *ActiveHard* conditions (*Mdn* = 47.95, *Z* = −3.340, <0.001, *r* = −0.159). Furthermore, in comparing the two Active conditions, viewers from this group also experienced significantly higher vmHRV during the *ActiveHard* condition, but with a small effect size: *Z* = −5.152, <0.001, *r* = −0.245.

LOW viewer’s HRV was significantly lower during *ActiveEasy* compared to *Passive* (*Mdn* = 31.38, *Z* = −6.071, <0.001), with a medium effect size of *r* = −0.301. Similarly, their HRV during *ActiveHard* (*Mdn* = 19.05) was significantly lower than for the *Passive* condition, (*Z* = −5.908, <0.001, *r* = −0.293). The different Active conditions did not elicit a significant difference in their psychophysiological states. Lastly, compared to the baseline vmHRV, viewers from the *HIGH* group showed higher vmHRV (*Mdn* = 35.00), whereas those from the *LOW* group showed lower vmHRV compared to the baseline (*Mdn* = 38.93). However, these differences were only significant for the *ActiveHard* condition (*HIGH*: *Z* = −8.667, <0.001, *r* = −0.430; *LOW*: *Z* = −1.853, 0.064, *r* = −0.092).

### 4.6. Physiological Synchrony

In exploring how closely viewers’ vmHRV synchronized with that of the gameplayers they were watching, we evaluated synchrony as a positive Spearman correlation coefficient representing the relationship between viewer vmHRV and gameplayer vmHRV for each pair. The *HIGH* pairs of viewers and gameplayers had moderately high positive correlations for *ActiveEasy*, *ActiveHard*, and *Passive* conditions at 0.55, 0.60, and 0.55, respectively. These were greater than the low positive correlation values between the LOW pairs, which were 0.33, 0.35, and 0.32, respectively. Figure 9 demonstrates these correlations. Upon further investigation into whether there were group differences, we performed a Mann-Whitney U test and found no significant differences between groups (*p* = 0.76, *p* = 0.11, and *p* = 0.07, respectively). Furthermore, according to a Wilcoxon signed-rank test, there were also no significant differences in synchrony between conditions.

### 4.7. Open-Ended Responses

The following section presents the qualitative findings derived from the written open-ended questions viewers encountered at the end of the study. We present the results here as emerging themes from a thematic analysis and complement the quantitative results, providing a contextual perspective of the similarities or differences in socioemotional and psychophysiological patterns. We determined two overarching themes of responses as being either socially oriented—guided by viewers’ understanding of others’ emotional states—or task-oriented—influenced by viewers’ engagement while watching the remote gameplay—and built sub-themes according to these for each question.

#### 4.7.1. Connecting Aspects of the Experiment

The socially-oriented aspect that helped viewers feel connected to gameplayers primarily involved the gameplayer’s success in the game (*N* = 9; P4, P8, P11, P12, P19, P20, and P23). Five viewers (P17, P18, P21, P22, and P24) also reported feeling close to gameplayers through vocalizing or hearing them, with one of these viewers specifically mentioning connecting through laughing together (“We both laughed at the mistakes” (P21)). Being able to see the first-person perspective of the gameplayers also helped several viewers feel connected to them (*N* = 3; P6, P14, and P22). Four viewers (P1, P3, P5, and P9) claimed task-oriented aspects, such as counting errors or the difficulty of the gameplay during the *ActiveHard* condition, helped them feel a sense of relatedness.

#### 4.7.2. Perceived Causes of Stress

The socially oriented stressors included witnessing the gameplayer making errors (e.g., hitting the ring against the wire) (*N* = 3; P4, P8, and P20) or the mere potential of the gameplayer to do so (*N* = 4; P7, P19, P21, and P22), and simply imagining the gameplayer’s stress (*N* = 1; P17). Of these eight participants who reported social stressors, two were from the *LOW* group (P4 and P20). Task-oriented stressors, such as keeping track of the errors and “maybe losing some counting” (P5), were only reported by the LOW viewers (*N* = 4; P1, P5, P10, and P15).

#### 4.7.3. Ideas for Improving Social Connectedness to the Gameplayer

The viewers’ suggestions to support social connectedness with gameplayers were primarily socially motivated. This was either as a live visual of them or as a figurative representation of them. Four viewers who did not use the communication feature still felt it would be beneficial. Of these viewers, three had low ratings of closeness during all conditions (P5, P12, and P15), while one only had low ratings during Active conditions (P11). Most viewers (N = 8; P1, P6, P7, P10, P14, P19, P22, and P23) felt a visual of the gameplayer could help, whether it was seeing them physically playing (“If I could see her move and have a visual of her as a person” (P7)) or a representation of their emotional reactions (e.g., “Some representations of the other person’s feelings/physiological state or of their face” (P19)). Lastly, two viewers who consistently reported *HIGH* Closeness across all conditions (P8 and P21) requested more communication modalities, although they did not specify what kind.

## 5. Discussion

In the following section, we discuss the role of feeling socially connected to a remote gameplayer during gameplay sessions and its effect on socioemotional and psychophysiological states.

### 5.1. The Interplay of Social Connectedness, Socioemotional Competences, and Physiological States While Watching Remote Gameplay

The experimental study presented in this paper demonstrates that viewers’ feelings of social connectedness to gameplayers during live gaming sessions relate significantly to how concerned they felt for the gameplayers, which is consistent with findings from other studies [21,22]. We also observed that viewers who felt more socially connected to the gameplayers experienced heightened parasympathetic activity during gaming sessions. Affective empathy, on the other hand, seemed to be unbiased by social connectedness and unrelated to parasympathetic activity. These findings suggest that parasympathetic activity is not contingent upon feelings of empathy during live gameplay sessions but rather on the perceived social connection to the gameplayer and concern for their well-being. This contradicts the aforementioned research, in which self-reported empathy correlated with increased vmHRV [24].

The limitations of social cues typically available during face-to-face interactions are commonly cited as challenging to virtual interactions [73,74]. In the absence of social cues, viewers who reported low closeness to gameplayers consistently expressed that communicating more with gameplayers would have been desirable, showing a motivation to connect. However, individuals who perceive or convey socioemotional cues even through indirect communication on virtual platforms can feel sufficiently connected to gameplayers [75,76]. For instance, hearing the gameplayer laughing and laughing along with them contributed to a sense of connectedness for viewers in our study, highlighting the importance of socioemotional cues to enhancing connectedness [77]. While the dialogue between participants was not recorded, we observed that viewers and gameplayers familiar with each other were generally inclined to engage in casual conversations during the *Passive* condition. Given that established relationships mediate feelings of closeness [37,78], if the majority of viewers in these pairs also felt high social closeness during this condition, we can also infer that these verbal interactions played a role in distinguishing empathic concern from viewers who felt little closeness to gameplayers during this condition.

### 5.2. The Effect of Gameplay Type on Social Connectedness, Empathic Concern, and the Psychophysiological States of Viewers

In this section, we answer the research question, “How does the type of gameplay affect the relationships between social connectedness, socioemotional competences, and psychophysiological states of remote viewers of gameplay?” Viewers reported higher levels of closeness during gameplay sessions compared to the unstructured sessions. While changes in viewers’ empathic concern ratings between gaming sessions corresponded with changes in vmHRV, the former did not change significantly. In this section, we attempt to explain these changes and their distinctions.

For the viewers in our study, the goal-oriented gaming sessions encouraged greater connectedness to streamers than the unstructured, passive gameplay session. However, there were distinctions in what contributed to feeling socially connected during these games, as evidenced in their comments. Viewers who felt little to no social connectedness were more concerned with a desire for the streamer to succeed, as explicitly mentioned in the following comment:

“The fact that I wanted them to succeed [made me feel connected]”(P11)

Here, the viewers’ investment in the streamer’s success contributed to their sense of gratification. While viewers who felt higher social connectedness with streamers were also concerned with the streamer’s success, sharing perspectives through a first-person view of the gameplay helped them feel closer to the streamer.

“The focus on the same specific points”(P6)

“The game, cursor, moving as [the streamer] moves”(P14)

“Being able to see what they see and hearing them talk”(P22)

Although viewers who felt socially connected to streamers expressed significantly greater empathic concern than those who felt low connectedness, these feelings of concern did not vary significantly between gameplay types, suggesting that the nature of the game is less relevant than the strength of the perceived social closeness to the streamer when it comes to fostering empathic concern.

Goal-oriented games provide a unique environment where shared objectives between players and the viewers foster a sense of collaboration and mutual involvement [79]. This dynamic may foster social connection by engaging the audience emotionally and cognitively in the players’ experiences. This was evident in the gameplay sessions for both the high and low socially connected viewer groups. The shared emotional investment in achieving common goals enhances empathy, as viewers relate to the players’ challenges and successes. The viewers who felt low social connectedness with streamers integrated the streamer’s success into their own reward system. Furthermore, in viewers who felt greater social connectedness to streamers during gaming sessions, this interdependence is exemplified as an integration of the streamer’s experience into their own experience, integrating the entire gameplay experience rather than solely the experience of succeeding and potentially increasing feelings of affiliation with the streamer [5]. Positive interdependence can also lead to prosocial behaviors [79]. These results indicate that watching goal-oriented games, particularly ones that encourage emotional investment in the streamer’s gameplay, may contribute to greater interpersonal closeness with streamers. In light of the importance of empathic concern in cultivating meaningful relationships and encouraging acts of kindness [6,7], supporting these experiences and fostering social connectedness may increase this socioemotional competence and lead to others in live-stream gaming communities.

#### Gameplay and Physiological States

When observing others in stressful scenarios, people experience heightened levels of psychological stress and anxiety, mirroring the emotional states of the others [80]. Our study demonstrated this through the elevated personal distress scores of viewers that matched their perception of the player’s elevated psychological stress during the challenging gameplay session. Furthermore, viewers who felt little connectedness to streamers also experienced heightened physiological stress, as evidenced by decreases in their vmHRV. However, the increased parasympathetic activity in viewers who felt more socially connected during this session suggests that physiological responses may not merely reflect their ability to empathize with streamers or the nature of gameplay. Instead, they may be more deeply linked with a sense of connectedness to the streamer and feelings of concern for their well-being.

We also sought to explore why socially connected viewers may have experienced heightened states of physiological relaxation while less socially connected viewers experienced more physiological stress, despite both groups reporting heightened levels of mental stress, and we examined the role of social connectedness in their open-ended responses. This provides insights into understanding the social facets of challenging, goal-oriented games that contribute to feelings of psychological stress and their sense of connectedness. The most common source of psychological stress reported by viewers in this study was related to partners making errors and experiencing failure. For instance, a viewer (P20) who felt little connectedness reported feeling stressed by “failed trials”. Indeed, “When the [gameplayer] makes errors” (P4) or “counting errors” (P15) were commonly mentioned stressors of viewers from this group. On the other hand, none of the comments made by viewers who felt high social connectedness involved observing the gameplayer making errors. Instead, a comment made by a viewer from this group expressed uncertainty:

“Watching [the gameplayer] play and not knowing if they were going to make mistakes.”(P22)

This uncertainty was similarly expressed by many viewers from this group, whether it was explicitly stated, such as

“Not sure if [the gameplayer] finishes”(P21)

or conveyed through an expectant statement, such as

“When [the gameplayer] got to the difficult sections”(P19)

The viewers in these comments seemed to infer what it feels like to fail and made predictions about what the streamer might feel like if they made errors or lost rather than actually witnessing them do so. The anticipatory tone of their reactions also aligns with the significantly heightened concern expressed by viewers in this group.

Consistent with existing research linking psychological stress to elevated physiological arousal [81], viewers with little to no social connectedness with streamers experienced decreases in vmHRV, which may be the vicarious psychological stress they felt through empathizing with the streamer as they make errors. However, despite also being able to empathize, feeling socially connected may have helped viewers from the other group re-evaluate the streamer’s experience of making errors. While errors can be simply defined as failure to achieve a reachable goal [82], the viewer’s perception of error also distinguishes their understanding of the streamer’s goals from their own interpretation of the outcomes [83]. This can help them evaluate the streamer’s experience and dynamically renegotiate their response. However, constructing an understanding of a streamer’s goals requires social cues from them, which, as previously mentioned, can be limited in virtual environments [73].

Occurrences of this are best exemplified in the increased parasympathetic arousal observed in viewers who felt higher degrees of closeness, which may be attributed to the parasympathetic activation associated with feeling concerned and access to social cues from the streamer [25,26]. Although this study did not capture audio between streamers and viewers, we informally noted that the streamers occasionally had audible reactions to making errors, such as laughter or expressions of amusement. These may have helped to alleviate the negative physiological responses of viewers to seeing errors being made by the streamer. The disconfirming of negative expectations led to calmer states [84]. In other words, the viewer’s apprehension of the negative feelings the streamer might have experienced when failing ultimately proved to be less unpleasant than initially anticipated when they heard the streamer responding in a positive way. This positive reappraisal of a tense scenario may have facilitated the decreased physiological stress observed in viewers who feel socially connected [85,86]. It is particularly revealing that these vmHRV states were significantly greater for these viewers during challenging gameplay, confirming that a strong sense of social connectedness may be linked to better emotional regulation and physiological well-being.

### 5.3. Psychophysiological Synchrony of Viewers and Streamers

In contrast to previous research, which found that individuals who empathized with another collocated person’s affective experience exhibited tightly coupled physiological responses (i.e., physiological synchrony) [43,44,87], viewers’ capacity for empathy towards gameplayers did not correspond with the differences in synchrony observed among the groups in our study. This suggests that empathy for a virtual person may be less of a significant factor in facilitating tightly coupled physiological interactions than feeling socially connected with them.

In this section, we attempt to explain the stronger vmHRV correlation (i.e., synchrony) between viewer-gameplayer pairs in which the viewers felt greater social connectedness to gameplayers, as well as the lack of significant differences in physiological synchronization between the conditions for these pairs. First, it is possible that these pairs exhibited greater physiological synchrony during gaming tasks due to the similar sense of social connectedness felt by the streamers, for whom the mutual responsibility for a shared objective can also strengthen feelings of connectedness [79]. As one gameplayer from our study put it,

“Knowing that we look at the same thing and I’m sort of responsible for our common success or fun in the game [made me feel connected with my partner]”(streamer partner of P6)

Next, it is worth reflecting on the level of physical activity gameplayers engage in during gaming tasks and its potential impact on physiological synchrony with viewers who were not required to engage in physical activity. The more challenging gameplay required streamers to move more slowly and cautiously, while the absence of constraints in the easier gameplay enabled them to move more quickly and freely. The restricted movements and parasympathetic activation associated with social connectedness may account for the higher physiological synchrony in the former task. In comparison, the unconstrained physical movements of the streamer during the easier gameplay may have elevated their heart rate enough to differentiate it from that of the viewer.

In order to clarify the increased physiological synchrony observed in the absence of gaming tasks, we reaffirm the tendency for pairs with viewers who reported a stronger sense of social connectedness to engage in light conversations. This interaction, as well as the similarities in emotional arousal and physical activity, may have contributed to greater synchronized physiological states during this condition [88]. Additionally, familiarity, which acted as a mediator for closeness in this particular scenario, may have influenced physiological synchrony. This finding is consistent with earlier studies suggesting that pre-existing relationships can evoke physiological synchrony [37,78].

We attribute the physiological synchrony between viewers and gameplayers during live stream gaming to social connectedness and concern for others’ well-being [88]. In streaming scenarios, social connectedness, shared affect, and more elaborate communication are the key contributors to physiological synchrony [37]. In either case, leveraging methods to reflect physiological synchrony to remote users could serve to validate and reinforce a sense of connectedness. This also holds the potential to facilitate the cultivation of social connectedness, a process that often requires significant time in virtual interactions [5].

## 6. Implications

The findings in this study demonstrate that viewers’ feelings of social connectedness to remote gameplayers are directly linked to increased empathic concern and improved physiological states, as indicated by heightened vagally mediated heart rate variability (vmHRV). This has several implications for understanding social connectedness, empathy, and physiological states during remote virtual interactions.

**Discernment of How Social Connectedness Enhances Empathy** Viewers were able to feel affective empathy, regardless of how socially connected they felt to their partners. However, empathic concern was mostly expressed by viewers who felt a stronger sense of connectedness to gameplayers, displaying higher levels of empathic concern. Integrating community-building tools, such as interactive discussion boards or discussions centered on shared emotional experiences, can further support social connectedness. These tools would help users reflect on one another’s experiences and foster deeper connections within their viewing communities.

**Understanding of the Relation between Social Connectedness and Psychophysiological Responses** Viewers who felt a stronger sense of connectedness to their remote gameplaying partners experienced greater vagally mediated heart rate variability (vmHRV), suggesting benefits to both emotional and physical health from such connections. Furthermore, this study identified physiological synchrony between remote gameplayers and viewers, particularly for pairs in which viewers felt higher social connectedness. This provides evidence of shared psychophysiological states even in remote, virtual environments.

Platforms could leverage physiological data to refine the user experience. For instance, real-time interaction tools, such as synchronized emoji reactions or live emotional overlays, could enhance users’ awareness of shared experiences. These features would allow viewers to feel more connected to others by making their reactions visible and aligned in real time. Additionally, the study’s insights into physiological synchrony can inform the development of group viewing enhancements. By analyzing synchronization patterns among users, platforms could suggest content that is more likely to resonate with groups or implement personalized settings that adapt playback based on physiological responses, thereby ensuring cohesive and engaging experiences for users.

**Exploration of the Role of Content Type** In our study, the goal-oriented gameplay fostered greater social connectedness compared to the *Passive* condition. The structured nature of such games appears to encourage socioemotional engagement compared to ambiguous situations. To maximize these benefits, streaming platforms or streamers could explore how to design experiences that emphasize collaborative goals and shared achievements, creating opportunities for viewers to feel actively engaged and socially connected. Such designs could amplify the positive effects of goal-oriented games on both social and physiological well-being.

**Contribution to Understanding Virtual Interactions** By addressing the dynamics of socioemotional competences and physiological responses in remote audiences, this study contributes to research on virtual social interactions. In particular, we highlight the importance of encouraging empathetic engagement during remotely streamed gameplay and its potential to enhance viewer well-being. For instance, offering features to improve communication between audience members and gameplayers, as well as first-person perspectives that contain more detailed information about the gameplayer, can enhance empathetic engagement.

## 7. Limitations and Future Work

This study offers valuable insights into the relationship between social connectedness, empathy, and physiological states in people watching remote virtual gameplay. However, given the controlled experimental nature of this study, there are several limitations that should be considered before extending the findings to real-world scenarios, such as live stream gaming. In this section, we address these limitations and provide areas for future exploration [8].

### 7.1. Physiological Indicators

While physiological metrics from BVP signals have been actively used as key indicators to explore psychophysiological states (particularly, pNN50 in this study), using multiple physiological measures together, including respiration [85,89], skin conductance [41,90], spontaneous eye-blinking [91], and vasomotor activities [43,92], may provide a holistic understanding of viewers’ socioemotional and physiological states. This would help further delve into the interplay between emotional and physical responses in future studies. Furthermore, although our study evaluated the signal quality of BVP signals through auto-correlation analysis, more recent studies have demonstrated the superior performance of neural network-based models [93,94,95], which can be used in future studies to optimize the selection of BVP signals for synchrony analysis.

### 7.2. Participant Sample and Representation

The study involved a relatively limited sample size (48 participants), limiting the generalizability of the findings. Additionally, the composition of the sample may limit the generalizability of the findings, as demographic diversity was not a primary focus of recruitment even though research has shown gender differences in streaming preferences [96]. Future studies should, therefore, aim to recruit larger and more diverse cohorts, encompassing a wide range of demographic variables such as age, gender, cultural background, and prior gaming experience. Such diversity would not only improve the robustness of the findings but also allow for a detailed exploration of how demographic characteristics interact with the observed relationships between social connectedness, empathy, and physiological responses.

In addition to increasing the number of participants and accounting for demographic differences, future research could explore alternative grouping criteria. While this study focused on grouping participants as strangers and acquaintances to examine the impact of prior relationships, grouping by demographic variables such as age, gender, or cultural background could reveal how these factors influence social dynamics and may provide further insights into the mechanisms underlying social connectedness.

In this study, we did not account for whether participants in the stranger group met in person before the experiment (e.g., in the lobby of the experimental lab while waiting for the study to begin). It could be very interesting to explore its impact on perceived connectedness in future work. In real-world live streaming scenarios, viewers typically do not meet streamers face-to-face. Future studies could explore entirely remote interactions to better reflect the nature of parasocial relationships in live streaming.

### 7.3. Sociability in Real-World Scenarios

This study was conducted in a controlled lab setting, constraining its ecological validity. In the context of live stream platforms, factors such as monetary donation and gifting systems, subscriptions, and the presence of other co-viewers all contribute to influencing social connectedness [97,98]. Live stream gaming streamers’ personalities, on-screen behavior, and physical appearance play crucial roles in fostering viewer engagement and connectedness. For instance, the attractiveness or personality of streamers enhances the viewer’s motivation to engage in streaming content [96]. Examining how these characteristics interact with viewer perceptions and physiological responses would be an important extension of this work.

Furthermore, the simplistic nature of the game fails to capture the complex mechanics and social intricacies of popular games that are typically live-streamed, such as embodiment, interactions with non-player characters, quest-solving, and decision-making, all of which can impact the level of closeness a viewer feels towards streamers. This simplified design was chosen to focus on fundamental mechanisms of how social connectedness relates to physiological changes. Future studies could explore more varied game genres to capture more realistic viewer experiences. Investigating genre-specific effects could provide deeper insights into how different types of games affect social connectedness and empathy, enhancing the relevance of this research to streaming platforms and gaming communities.

Although this research involved the manipulation of variables, it would be intriguing to investigate these and other intricate socioemotional dynamics that occur in real-world live-stream gaming further. For instance, another limitation of the current study is its cross-sectional design, which captures only a snapshot of social connections. Future research should consider adopting a longitudinal approach to understand how these connections evolve over time (e.g., diary study). Such an approach would provide richer insights into the dynamics of social inter-relations and their influence on the reported associations. Incorporating temporal dimensions could significantly strengthen contributions to the field and enhance our understanding of the developmental aspects of social connections.

## 8. Conclusions

Inspired by the prominence of live-stream gaming as a means of cultivating social connections, the research presented in this paper highlights the considerable relationship between feeling socially connected and experiencing socioemotional competences and beneficial physiological states. In particular, the study demonstrates that viewers who feel more connected to virtual partners also care more for their well-being and experience calmer physiological states, ultimately contributing to improved overall well-being. Furthermore, the study emphasizes the role of gameplay type in modulating these effects, as goal-oriented gameplay appears to encourage increased socioemotional responses. Additionally, the observed physiological synchrony between viewers and remote virtual partners with whom they feel a connection highlights the significance of parasocial relationships in fostering stronger social bonds. This study offers new insights into the benefits of social connectedness formed through watching remote gameplay. Focusing on how empathy and empathic concern during streamed remote gameplay relate to viewers’ physical health and how gameplay type influences these relationships highlights the potential for designing live-stream experiences that enhance socioemotional connections, suggesting that such designs could counteract the negative trends in social competences and foster more supportive, connected communities.

## Figures and Tables

**Figure 1 sensors-25-00872-f001:**
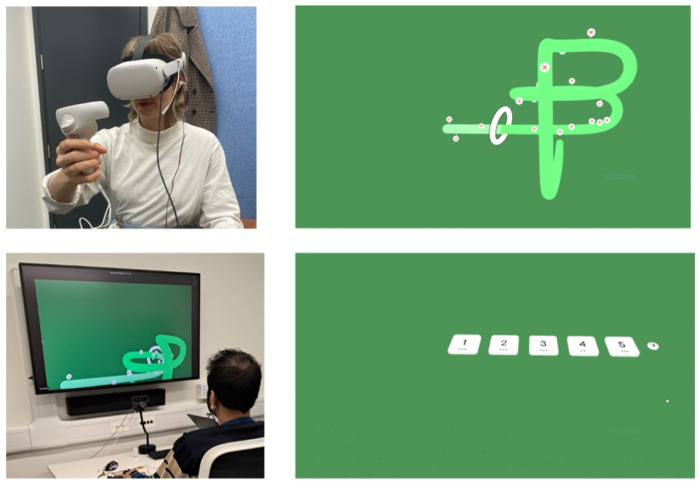
A pair of study participants and screenshots of gameplay. (**top left**) Gameplayer setup; (**top right**) screenshot of the *ActiveHard* task which illustrates errors as red crosses in white circles; (**bottom left**) viewer setup; **(bottom right**) screenshot of the *Passive* task.

**Figure 2 sensors-25-00872-f002:**
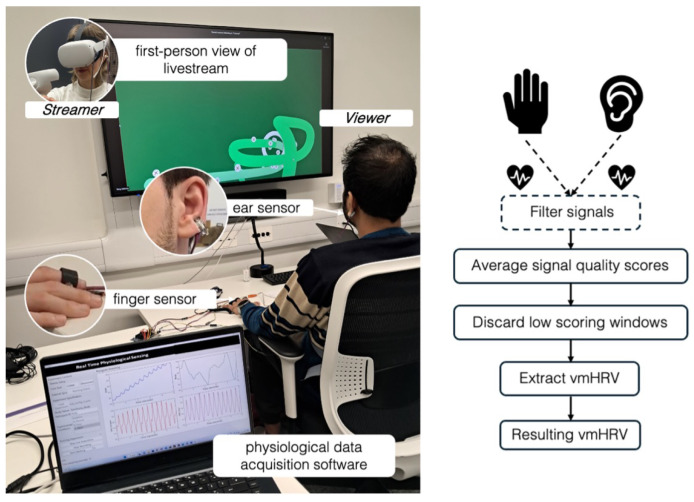
(**left**) Experimental setup for viewer participant wearing physiological sensors. (**right**) Overview of physiological data collection.

**Figure 3 sensors-25-00872-f003:**
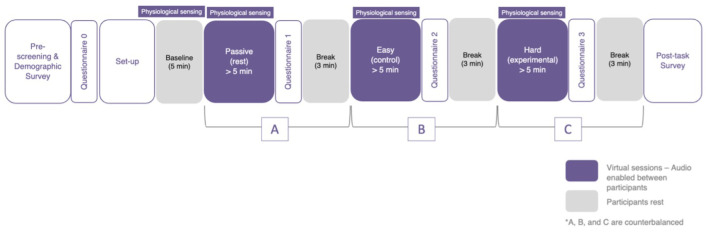
Diagram of the study protocol (* denotes counterbalanced conditions).

**Figure 4 sensors-25-00872-f004:**
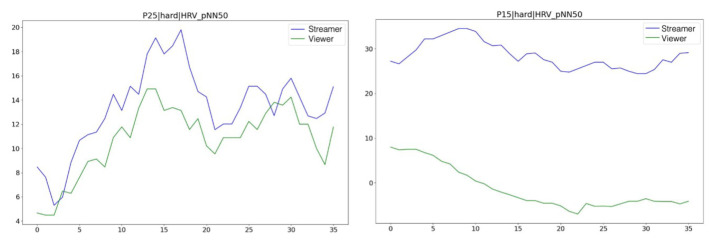
Examples of vmHRV signals of gameplayer-viewer pairs during the *ActiveHard* condition. (**left**) A pair from the *HIGH* social connectedness group shows synchrony between gameplayers and viewers. (**right**) A pair from the *LOW* social connectedness group where synchrony between gameplayers and viewers is not observed.

**Figure 5 sensors-25-00872-f005:**
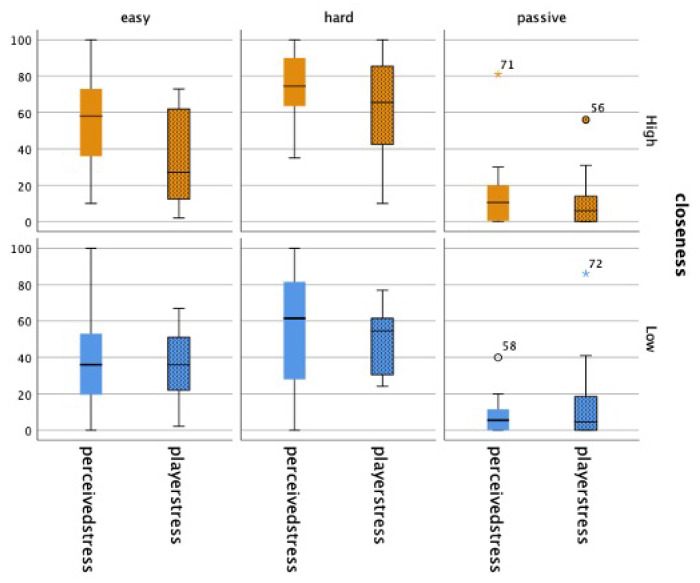
Viewer’s perception of psychological stress felt by gameplayers (“perceivedstress”) and actual psychological stress reported by gameplayers (“playerstress”). “playerstress” acts as a reference for each condition (* denotes outliers).

**Figure 6 sensors-25-00872-f006:**
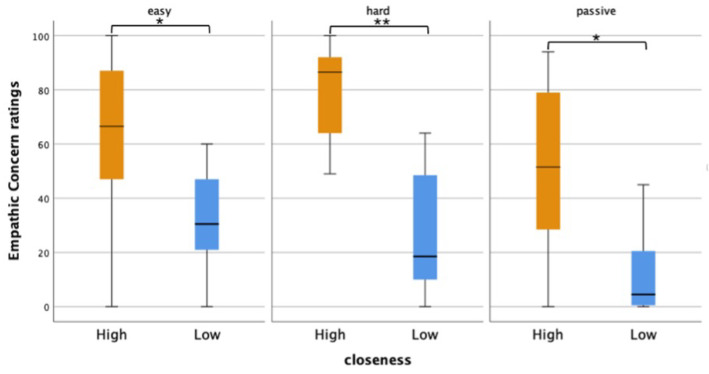
Empathic Concern ratings for the HIGH and LOW closeness groups across experimental conditions. Significant between-group differences are marked with asterisks (* Bonferroni < 0.017, ** <0.001).

**Figure 7 sensors-25-00872-f007:**
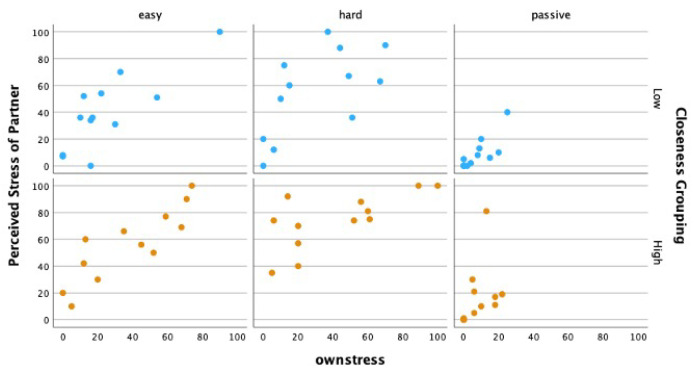
Affective empathy of viewers as a correlation between their perception of the gameplayer’s felt stress and their own felt stress for LOW and HIGH groups.

**Figure 8 sensors-25-00872-f008:**
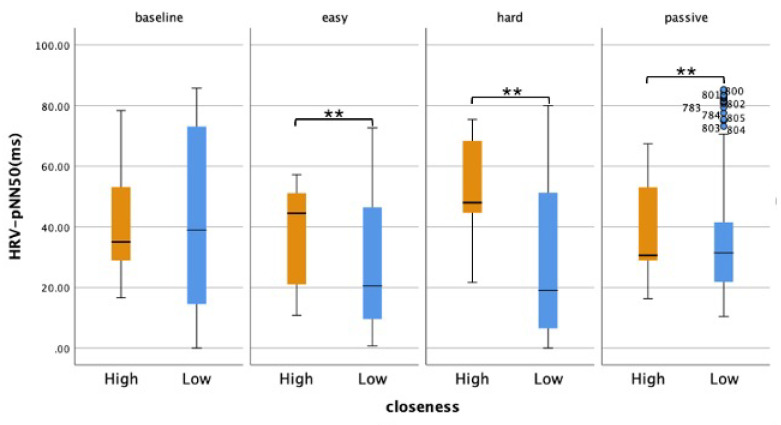
vmHRV across the conditions for the HIGH and LOW Closeness groups. Significant between-group differences are marked with asterisks (** <0.001).

**Figure 9 sensors-25-00872-f009:**
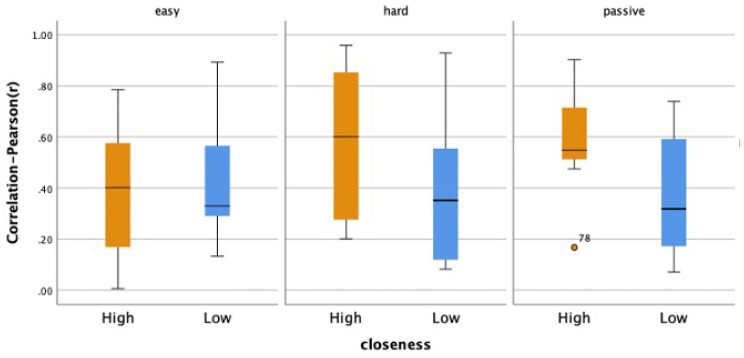
vmHRV correlation across conditions for the HIGH and LOW Closeness groups.

**Table 1 sensors-25-00872-t001:** Pair groupings based on viewers’ social connectedness ratings.

Pair	Relationship	Baseline	*ActiveEasy*	*ActiveHard*	Passive
Median Score		40.0	61.5	56.5	21
1	None	low	low	low	low
2	None	low	high	high	low
3	None	low	high	high	low
4	None	low	low	low	low
5	None	high	low	low	low
6	Romantic partners	high	high	high	high
7	None	low	high	high	low
8	Romantic partners	high	high	high	high
9	None	low	low	low	low
10	Romantic partners	low	low	low	low
11	None	low	low	low	high
12	None	low	low	low	low
14	Romantic partners	high	high	high	high
15	None	low	low	low	low
16	Friends	high	low	low	high
17	Romantic partners	high	high	high	high
18	Friends	high	high	low	high
19	Friends	high	high	high	high
20	None	low	low	low	low
21	Romantic partners	high	high	high	high
22	Romantic partners	high	high	high	high
23	Romantic partners	high	high	high	high
24	Romantic partners	high	high	high	high
25	Romantic partners	high	low	high	low

**Table 2 sensors-25-00872-t002:** Correlation values representing AffectiveEmpathy as the relationship between viewer perception of streamer stress (PerceivedStress) and their own stress ratings across experimental conditions for the HIGH and LOW Closeness groups.

	HIGH			LOW	
	*r*	*p*		*r*	*p*
*ActiveEasy*	0.865	<0.001 **		0.806	0.002 *
*ActiveHard*	0.696	0.012 *		0.621	0.031 *
Passive	0.398	0.200		0.801	0.002 *

Statistically significant results are marked with asterisks (* <0.05, ** <0.001).

**Table 3 sensors-25-00872-t003:** The results of a Wilcoxon signed-rank test comparing HRV between different conditions for the HIGH and LOW Closeness groups.

	HIGH			LOW	
Condition Comparison	* Z *	* p *		* Z *	* p *
*Baseline-ActiveEasy*	−0.971	0.331		−1.107	0.268
*Baseline-ActiveHard*	−8.667	<0.001 **		−1.853	0.064
*Baseline-Passive*	−6.894	<0.001 **		−5.064	<0.001 **
*Passive-ActiveEasy*	−4.075	<0.001 **		−6.071	<0.001 **
*Passive-ActiveHard*	−3.340	<0.001 **		−5.908	<0.001 **
*ActiveEasy-ActiveHard*	−5.152	<0.001 **		−1.714	0.087

Statistically significant results are marked with asterisks (** <0.001).

## Data Availability

The original contributions presented in this study are included in the article. Further inquiries can be directed to the corresponding author.

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
