# Peer review of "Streaming to Connect: Exploring How Social Connectedness Relates to Empathy Types and Physiological States in Remote Virtual Audiences"

_sensors, 2025, doi:10.3390/s25030872_

Round 1

Reviewer 1 Report

Comments and Suggestions for Authors

1.This article observed the psychological activities of the audience towards social connectivity and empathy through remote gaming experiments, and analyzed the importance of social connectivity in influencing empathy. It has certain innovation and practical significance, but this article still needs further revision.

2.Is it necessary to provide a quantitative description of the experiment in the abstract? If there are relevant conclusions, can they be elaborated in the abstract.

3.The COVID-19 pandemic has further increased people's neglect of empathy, specifically referring to their neglect of empathy? Or is it the neglect of the COVID-19 pandemic? Is the expression here appropriate?

4.In the experiment, participants were divided into two groups: strangers and acquaintances. Is there any other way of grouping, such as by age, gender, or other criteria?

5. Does the experimental grouping directly affect the conclusion, or how does this method effectively support the conclusion of the paper?

6. Goal oriented games can better promote a sense of social connection between the audience and players, thereby enhancing empathy and physical health for the audience. Can this be further explained?

Author Response

Comments 1: This article observed the psychological activities of the audience towards social connectivity and empathy through remote gaming experiments, and analyzed the importance of social connectivity in influencing empathy. It has certain innovation and practical significance, but this article still needs further revision.

Response 1: We sincerely thank the reviewer for their thoughtful feedback and for recognizing the innovation and practical significance of our study. We appreciate the reviewer's constructive feedback and valuable suggestions, which have helped us improve the quality of the paper.

Comments 2: Is it necessary to provide a quantitative description of the experiment in the abstract? If there are relevant conclusions, can they be elaborated in the abstract.

Response 2: Thank you for this comment. While we did not provide quantitative descriptions, we elaborated on relevant conclusions by emphasizing the broader implications of the findings. 
These revisions can be found on page 1, lines 9-12:
"The results revealed that viewers who felt greater social connectedness to remote partners also felt more empathic concern and had significantly heightened vmHRV across all conditions compared to those who felt lower connectedness."

Comments 3: The COVID-19 pandemic has further increased people's neglect of empathy, specifically referring to their neglect of empathy? Or is it the neglect of the COVID-19 pandemic? Is the expression here appropriate?

Response 3: Thank you for pointing this out. We agree that there is a bit of confusion around this sentence and have revised the phrasing to clarify that the pandemic has influenced a societal decline in empathy due to its social and emotional repercussions.
These revisions can be found on page 1, lines 32-34:
"Socioemotional challenges brought about by the circumstances of the COVID-19 pandemic have had repercussions on declining empathy. "

We have also added two references to support this:

van de Groep, S., Zanolie, K., Green, K. H., Sweijen, S. W., \& Crone, E. A. (2020). A daily diary study on adolescents’ mood, empathy, and prosocial behavior during the COVID-19 pandemic. \textit{PloS one}, \textit{15}(10), e0240349.

Saladino, V., Algeri, D., \& Auriemma, V. (2020). The psychological and social impact of Covid-19: new perspectives of well-being. \textit{Frontiers in psychology}, \textit{11}, 577684.

Comments 4: In the experiment, participants were divided into two groups: strangers and acquaintances. Is there any other way of grouping, such as by age, gender, or other criteria?

Response 4: We appreciate this suggestion. The categorization of participants as strangers or acquaintances served to facilitate the investigation of differences between participants who felt more close with their partners and those who felt less close. We provided a brief explanation in the Methods section why grouping by "strangers" and "acquaintances" was central to the research question, as in our Response 3. 

Nonetheless, we find the suggestion valuable, now suggesting the inclusion of alternative grouping criteria in the "Limitations and Future Work" section. These include demographic factors (age, gender, cultural background) alongside an explanation of the current grouping rationale and an acknowledgement of opportunities for more nuanced insights in future research.
These revisions can be found on page 20, lines 722-727:
"In addition to increasing the number of participants and accounting for demographic differences, future research could explore alternative grouping criteria. While this study focused on grouping participants as strangers and acquaintances to examine the impact of prior relationships, grouping by demographic variables such as age, gender, or cultural background could reveal how these factors influence social dynamics and may provide further insights into the mechanisms underlying social connectedness."

Comments 5: Does the experimental grouping directly affect the conclusion, or how does this method effectively support the conclusion of the paper?

Response 5: Thank you for raising this question. We provided a brief explanation in the Methods section why grouping by "strangers" and "acquaintances" was central to the research question, as in our Response 3. Since the study aims to explore the relationship of closeness with other variables, the experimental grouping was important to distinguish two separate groups.
These revisions can be found on page 4, lines 176-177:
"This distinction was crucial to ensure that one group had a higher closeness to intimate relationships."

Comments 6: Goal oriented games can better promote a sense of social connection between the audience and players, thereby enhancing empathy and physical health for the audience. Can this be further explained?

Response 6: We agree with this comment and have expanded the explanation of how goal-oriented games promote social connections, empathy, and physical health in the Discussion and Implications sections. The updated text will clarify that the perception of shared goals foster collaboration and mutual involvement between players and the audience, strengthening social bonds. We also explain how observing players' challenges and successes enhances empathy by aligning the audience with their emotional experience. 
These revisions can be found on page 17, lines 522-539 and page 20, lines 680-684:
"Goal-oriented games provide a unique environment where shared objectives between players and viewers foster a sense of collaboration and mutual involvement [Johnson, 2010]. This dynamic may foster social connection by engaging the audience emotionally and cognitively in the players' experiences. This was evident in the gameplay sessions for both groups of high and low socially connected viewers. The shared emotional investment in achieving common goals enhances empathy, as viewers relate to the players' challenges and successes. "
"To maximize these benefits, streaming platforms or streamers could explore how to design experiences that emphasize collaborative goals and shared achievements, creating opportunities for viewers to feel actively engaged and socially connected. Such designs could amplify the positive effects of goal-oriented games on both social and physiological well-being."

Reviewer 2 Report

Comments and Suggestions for Authors

Review for sensors-3412122

This is a substantial overall manuscript with multiple areas where content could improve or strengthen its contribution. First, the authors could explain their rationale for using vmHRV as the primary physiological measure. While the inclusion of vmHRV has a strong basis in the existing literature, it would strengthen the paper and the discussion if other potential physiology, such as skin conductance, respiration rate, or face temperature as measures, were given justification for why they are not included. Primarily, multiple measures such as physiological responses would have given a more holistic view of how the viewers were feeling both physically and emotionally.

1) This experiment could be improved by using more realistic streaming settings. A relatively simple loop-the-wire game like that in use does not seem to reflect the richness and complexity of normal streaming practice. Expanding to include more complex games or more varied streaming content would help connect the dots between lab findings and real-life applications. Furthermore, the authors could investigate how social connectedness and physiological mechanisms may differ based on varied genres of play.

Sample size and composition represent another domain for improvement. Some analyses may be limited by the statistical power of having only 48 participants. A broader and more diverse cohort of participants would increase the generalizability of the findings. In addition, the paper could benefit from a more detailed exploration of how demographic characteristics (e.g., age, gender, previous gaming experience) would interact with the observed relationships between social connectedness and physiological responses.

The signal processing procedures used to extract useful features (e.g., most stable point, standard deviation, changes over time) from the physiological data may lack detail, weakening the methodology section. Although the general procedures were provided, a more detailed approach to handling motion artefacts and sustaining signal quality is needed for others to replicate the analysis. Further, the authors might explain the threshold for excluding data points and addressing missing data.

The section regarding practical implications can be elaborated on. Although the authors briefly discuss possible implications for streaming platform design, they might elaborate further on the specific ways their findings could inform real-world practice. This may be in the form of explicit suggestions for features to better support social connectedness or specific ways to leverage physiological synchrony within streaming platforms.

Further, it could have engaged more with the temporal dimensions of the development of social connections. The current study represents a snapshot of social connections; however, we would gain better insights by knowing how these connections shift over time. A longitudinal aspect or some commentary on how inter-relations might affect the reported associations would strengthen the paper's contribution to the field.

Overall, these recommendations can build on what is already an impressive and informative study and ensure that the research's academic and practical implications are capitalised on as much as possible.

Author Response

Comment 1: The authors could explain their rationale for using vmHRV as the primary physiological measure. While the inclusion of vmHRV has a strong basis in the existing literature, it would strengthen the paper and the discussion if other potential physiology, such as skin conductance, respiration rate, or face temperature as measures, were given justification for why they are not included. Primarily, multiple measures such as physiological responses would have given a more holistic view of how the viewers were feeling both physically and emotionally.

Response 1: Thank you for this suggestion. We addressed this review by clarifying the rationale for using vmHRV and as the primary physiological measure over other types of measures by emphasizing the practical constraints of the VR environment. While we agree that other measures, such as skin conductance and respiration rate, could have provided a more holistic views, we originally highlighted in the discussion that they were excluded in this study due to the fact that the VR play settings involve various hand motions causing motion artefacts in physiological sensing e.g., GSR from fingers or BVP from fingers and wrists). This is crucial as such artefacts lead to unreliable physiological measurements and inaccurate conclusions [Joshi et al., 2023]. 
These revisions can be found on page 8, lines 202-210:
"The primary physiological measure in this study was vagally mediated heart rate variability (vmHRV), which was derived using ear photoplethysmography (PPG) sensors. The ear PPG sensor was specifically chosen due to its minimal interference with participant movements in the VR environment, avoiding common issues such as motion artifacts that can arise from hand- or wrist-worn devices. Other physiological measures, such as skin conductance, respiration rate, or face temperature, were excluded as they typically require sensor placements that could interfere with the gameplayer's natural use of VR controllers and often cause inaccurate measurements [Joshi et al., 2023]. We found this setup to be the least intrusive way to collect physiological data while preserving the ecological validity of the VR experience."

Comment 2: This experiment could be improved by using more realistic streaming settings. A relatively simple loop-the-wire game like that in use does not seem to reflect the richness and complexity of normal streaming practice. Expanding to include more complex games or more varied streaming content would help connect the dots between lab findings and real-life applications. Furthermore, the authors could investigate how social connectedness and physiological mechanisms may differ based on varied genres of play. 

Response 2: Thank you for highlighting this point. We completely agree that the simplicity of the game used does not reflect those typically played in live stream gaming. We addressed this review by elaborating on the limitations of the simplified game design and suggesting future research to incorporate more complex and varied streaming scenarios. Based on the reviewer's suggestion, the updated text also proposed exploring different genres of games to better reflect real-world streaming practices. 
These revisions can be found on page 21, lines 747-753:
"While this simplified design was chosen to focus on fundamental mechanisms of how social connectedness relates to physiological changes. Future studies could explore more varied game genres to capture more realistic viewer experiences. Investigating genre-specific effects could provide deeper insights into how different types of games affect social connectedness and empathy, enhancing the relevance of this research for streaming platforms and gaming communities."

Comment 3: Sample size and composition represent another domain for improvement. Some analyses may be limited by the statistical power of having only 48 participants. A broader and more diverse cohort of participants would increase the generalizability of the findings.
In addition, the paper could benefit from a more detailed exploration of how demographic characteristics (e.g., age, gender, previous gaming experience) would interact with the observed relationships between social connectedness and physiological responses.

Response 3: We agree that the limitations of the sample size and composition leave room for improvement. Reflecting on this, we have now elaborated on the limitations of the small sample size and lack of diversity in the participant cohort. Based on the reviewer's comment, we proposed increasing the sample size and recruiting participants from varied demographic backgrounds to improve generalizability. Additionally, we recommended exploring how demographic characteristics (e.g., age, gender, gaming experience) interact with social connectedness and physiological responses.
These revisions can be found on page 20, lines 713-721:
"Additionally, the composition of the sample may limit the generalizability of the findings, as demographic diversity was not a primary focus of recruitment even though research has shown gender differences in streaming preferences [Yu et al., 2022]. Future studies should, therefore, aim to recruit larger and more diverse cohorts, encompassing a wide range of demographic variables such as age, gender, cultural background, and prior gaming experience. Such diversity would not only improve the robustness of the findings but also allow for detailed exploration of how demographic characteristics interact with with the observed relationships between social connectedness, empathy and physiological responses."

Comment 4: The signal processing procedures used to extract useful features (e.g., most stable point, standard deviation, changes over time) from the physiological data may lack detail, weakening the methodology section. Although the general procedures were provided, a more detailed approach to handling motion artefacts and sustaining signal quality is needed for others to replicate the analysis. Further, the authors might explain the threshold for excluding data points and addressing missing data.

Response 4: We thank the reviewer for highlighting the need for additional details. We have now addressed this by elaborating both feature extraction and handling of motion artifacts. 

The revisions can be found on page 8-9, lines 301-329. We provide a brief overview of the descriptions added here: 

Signal Quality: The periodic nature of BVP signals in healthy individuals allows the use of auto-correlation to identify motion artifacts, which show reduced auto-correlation values [Pradhan et al., 2017]. Signal quality is assessed by the maximum autocorrelation amplitude in a normalized correlogram [Pradhan et al., 2017]. To achieve this, normalized correlograms were computed every 0.5 seconds over 2.5-second intervals.

Feature Extraction: pNN50, a vmHRV metric, was calculated as the percentage of successive heartbeat intervals differing by over 50 ms. A 120 s window with a 5 s step was used. Segments with average signal quality scores below 0.7 (derived empirically) were discarded [Zanon et al., 2020].

Missing Data: If a 120s window was discarded for one participant, their partner's corresponding window was also discarded to maintain balanced analysis. Physiological synchrony using pNN50 correlation was calculated only for high-quality time-matched segments. If there were fewer than three such segments, the entire session for the pair was excluded from analysis.

Comment 5: The section regarding practical implications can be elaborated on. Although the authors briefly discuss possible implications for streaming platform design, they might elaborate further on the specific ways their findings could inform real-world practice. This may be in the form of explicit suggestions for features to better support social connectedness or specific ways to leverage physiological synchrony within streaming platforms.

Response 5: Thank you for this suggestion. We expanded the "Implications" section to elaborate on practical implications for streaming platform design based on the study's findings. These suggestions include integrating real-time interaction tools, enhancing group viewing experiences through personalized physiological synchronization, and refining recommendations using physiological data. Additionally, we proposed incorporating community-building tools such as discussion boards centered on emotional experiences to further foster empathy and social connectedness.
These revisions can be found on page 19, lines 656-660:
"Integrating community-building tools, such as interactive discussion boards or discussions centered on shared emotional experiences, can further support social connectedness. These tools would help users reflect on their one another's experiences and foster deeper connections within their viewing communities."

Comment 6: The paper could have engaged more with the temporal dimensions of the development of social connections. The current study represents a snapshot of social connections; however, we would gain better insights by knowing how these connections shift over time. A longitudinal aspect or some commentary on how inter-relations might affect the reported associations would strengthen the paper's contribution to the field.

Response 6: We appreciate this suggestion and have revised the text in the "Limitations and Future Work" section emphasizing the limitation of the study's cross-sectional design. The addition encourages future research to adopt longitudinal approaches to better understand how social connections develop and evolve over time. Specifically, we suggest that integrating temporal dynamics and longitudinal studies would enhance the study's contribution by providing deeper insights into the developmental aspects of social interrelations.
These revisions can be found on page 21, lines 757-762:
"Future research should consider adopting a longitudinal approach to understand how these connections evolve over time (e.g. diary study). Such an approach would provide richer insights into the dynamics of social interrelations and their influence on the reported associations. Incorporating temporal dimensions could significantly strengthen the contribution to the field and enhance our understanding of the developmental aspects of social connections."

Round 2

Reviewer 2 Report

Comments and Suggestions for Authors

I am happy with the revisions. However, the methodology section can still be improved

Author Response

Comments: I am happy with the revisions. However, the methodology section can still be improved

Response: 

We greatly appreciate the reviewer's time and consideration in helping us improve the quality of our paper. In response to their feedback, we have made the following revisions:

To emphasize that our methods were inspired by previous research, we have added this sentence to page 4, lines 160-164:
"By building upon these studies, our research extends the application of vmHRV measurement to remote virtual audience settings. This adaptation of established methodologies allows us to explore how social connectedness and empathy manifest in digital interactions, providing valuable insights for designing virtual platforms that promote socioemotional well-being."

To further strengthen the rationale behind our study design, we have added the following introductory sentences to the methodology section on page 4, lines 177-183:
"Our study design leverages the established relationship between vmHRV and socioemotional competences to explore these dynamics in a novel context. Building on evidence from prior research (e.g., (Lischke et al., 2018, Bagnis et al., 2024), we employed continuous measurement of vmHRV using a PPG sensor to objectively assess participants’ physiological states alongside measure of empathy during remote gaming sessions. We adapted our environmental setup and approach from a previous study on physiological measurements and empathy evaluations during video viewing (Cho et al., 2023). The following subsections detail our methodology."

Bagnis, A., Ottaviani, C., \& Mattarozzi, K. (2024). Face your heart: resting vagally mediated Heart Rate Variability Shapes Social Attributions from facial appearance. \textit{Current Psychology}, \textit{43}(2), 1855-1863.

Cho, A., Park, S., Lee, H., \& Whang, M. (2023). The physiological measurement and evaluation of empathy of video content. \textit{Scientific Reports}, \textit{13}(1), 20190.

Lischke, A., Pahnke, R., Mau-Moeller, A., Behrens, M., Grabe, H. J., Freyberger, H. J., ... \& Weippert, M. (2018). Inter-individual differences in heart rate variability are associated with inter-individual differences in empathy and alexithymia. \textit{Frontiers in psychology}, \textit{9}, 229.